# Effect of HA Content on Microstructure and Properties of Ti-27Nb-17Ta-8Zr/HA Composite

**DOI:** 10.3390/ma16145095

**Published:** 2023-07-19

**Authors:** Qinggong Jia, Shuhua Liang, Qingxiang Wang

**Affiliations:** 1School of Material Science and Engineering, Xi’an University of Technology, Xi’an 710048, China; jiacrack@163.com; 2Xi’an Juneng Engineering Medicine Technology Co., Ltd., Xi’an 710026, China; 3Sino-Euro Materials Technologies of Xi’an Co., Ltd., Xi’an 710018, China

**Keywords:** Ti-27Nb-17Ta-8Zr/HA composites, spark plasma sintering, microstructure, mechanical properties, in vitro biocompatibility

## Abstract

In this paper, Ti-27Nb-17Ta-8Zr/HA series composite materials were prepared by spark plasma sintering (SPS) technology. The medical titanium alloy (Ti-27Nb-17Ta-8Zr) with good mechanical properties, wear resistance, and corrosion resistance was combined with the hydroxyapatite (HA) bioactive ceramic with high biological activity and bone-binding ability. Moreover, the density, microstructure evolution, metal/ceramic reaction, mechanical behavior, in vitro bioactivity, and influencing mechanisms of composite materials with different HA contents were studied. The research results indicate that all biological composite materials are composed of β-Ti solution, α-Ti, and ceramic phases (T_i2_O, CaTiO_3_, CaO, Ti_x_P_y_). With the increase of HA content, the compressive strength and yield strength of the composite material show a trend of first increasing, then decreasing, and then slowly increasing. After soaking in SBF artificial simulated body fluid for 5 days, the deposition of elements such as Ca and P on the surface significantly increased, while elements such as Ti, Nb, Ta, and Zr were evenly distributed in the matrix, demonstrating good in vitro mineralization ability and facilitating the attachment and growth of osteoblasts.

## 1. Introduction

Titanium-based hydroxyapatite (HA) biocomposites not only have high specific strength, low elastic modulus, good wear resistance, corrosion resistance, and biocompatibility of titanium alloys, but they also have excellent osteo-inductive properties and bioactivity of hydroxyapatite (HA), which can promote the osseo-integration between materials and human tissues, becoming one of the research focuses of biomedical implant materials [1,2,3,4,5,6,7]. Balbinotti et al. [8] prepared the Ti/HA biocomposites by powder metallurgy with different particle sizes of HA and Ti powder as raw materials. The study showed that powder particle size was the main factor affecting sintering effect and mechanical strength. The results after soaking in SBF indicate that the surface-generated Ca-P deposition of Ti/HA biocomposites prepared with fine particles or nanoscale powders is more uniform. In addition, HA decomposes during the sintering process at a temperature of approximately 1026 °C. This results in the formation of CaTiO_3_, Ti_x_P_y_, and TCP phases, which aggregate at the boundaries of Ti particles, leading to intergranular fracture and a decrease in the mechanical properties of the material. Buciumeanu M et al. [9] prepared Ti-6Al-4V/HA composites with 5–15% HA content by hot pressing. The results showed that the HA content played an important role in corrosion resistance in artificial saliva at 37 °C. With the increase of HA, the wear resistance of samples was greater, and the corrosion tendency was relatively low. The wear rate was lower than that of Ti-6Al-4V alloy. S H Park et al. [10] prepared Ti-35Nb-7Zr/10 HA composite materials by discharge plasma sintering technology using Ti, Nb, and Zr powders, as well as HA powders as raw materials after ball milling at different times. The research results showed that HA reacted during the sintering process to generate Ti_2_O, CaO, Ti_x_P_y_, CaTiO_3_, and CaZrO_3_ phases, and its microhardness gradually increased with the prolongation of ball milling time and the increase of HA content. The biocompatibility of Ti-35Nb-7Zr alloy improved with the addition of HA, but its sintering performance decreased. Despite the success of composites and technologies, these composites still have serious weakness. For example, the elastic moduli of those composites (Ti/HA: 102.6 GPa and Ti6Al4V/HA: 51.11 GPa) are much higher than that of human natural bone (10–40 GPa). The great difference of elastic moduli between implant materials and natural bone will cause a stress-shielding effect, bone resorption, implant loosening, and even implantation failure. Moreover, some techniques, such as PEO, CVD, and sol-gel, are also used to prepare the Titanium-based hydroxyapatite (HA) biocomposites [11,12,13]. However, those methods require a lot of pre-treatment work, and there are few reports on directly preparing composite materials using HA and titanium alloy powder through spark plasma sintering technology.

Spark plasma sintering technology has the advantage of having a low sintering temperature, short sintering time, high density, clean preparation process, and effective prevention of HA decomposition during high-temperature sintering [14,15]. It has become one of the effective methods for preparing active ceramic-reinforced titanium matrix composites [16]. Therefore, based on the previous research work, HA active ceramics were selected as the reinforcement phase and Ti-27Nb-17Ta-8Zr alloy as the matrix phase to prepare Ti-27Nb-17Ta-8Zr/HA biocomposites using SPS technology. In order to better study the effect of HA on composite materials, different amounts of HA were studied. Moreover, the effects and mechanisms of different HA contents on the microstructure, evolution, mechanical properties, and in vitro mineralization properties of the composites were studied.

## 2. Experimental Section

The experimental material is pure Ti powder (purity 99 5%, average particle size 125 μm), Nb powder (purity 99.9%, average particle size 75 μm), Ta powder (purity 99.5%, average particle size 5 μm), Zr powder (purity 99.5%, average particle size 5 μm) (Northwest Nonferrous Metals Research Institute), and HA powder (rod-shaped, with an average diameter of 75 nm) (Chengdu Kelon Chemical Reagent Factory, Chengdu, China). The preparation process of composite materials is as follows: a certain proportion of Ti powder, Nb powder, Ta powder, and Zr powder are placed in a V-shaped mixer and mixed with high-purity Ar gas for 2 h. The SEM photos and energy spectrum analysis after mixing are shown in Figure 1a,b, from which it can be seen that Ta and Zr are uniformly distributed on the surface of spherical Ti and Nb powders. Weigh the self-made rod-shaped HA powder as shown in Figure 1c,d, according to the mass fraction of 2%, 4%, 6%, 8%, and 10%, with the remaining amount as the mixed powder. Continue to put it into a V-shaped mixer (JHT-5, Zhengzhou Jinhe Technology Co., Ltd., Zhengzhou, China), mix it with high-purity Ar gas for 2 h with a speed of 30 r/min to obtain Ti-27Nb-17Ta-8Zr/HA mixed powder. Place the composite powder into a graphite mold, vacuum it to 2 Pa, apply 50 MPa axial pressure, and then sinter it on the SPS sintering system (LABOX-650F, SINTERLAND, Nagaoka, Japan). Heat it up to 1200 ℃ at a heating rate of 150 °C/min. After holding at 1200 °C for 10 min and cooling in the furnace to room temperature, a Ti-27Nb-17Ta-8Zr/HA composite material sample was obtained. The process curve of SPS sintering is shown in Figure 2. The currents and voltages are 1500 A and 10 V, respectively. The graphite is used as a mold for sintering.

The calculation of relative density from theoretical values for alloy samples was carried out according to the formula (*RD*):(1)$$RD(\%)=\frac{100}{\frac{\omega_{1}}{\rho_{1}}+\frac{\omega_{2}}{\rho_{2}}}$$
where *ω* is the component weight content, *ρ*-theoretical component density.

XRD-7000 X-ray diffraction (XRD) was used to analyze the phase structure of the sintered composite material sample. The emission wavelength of Cu Kα was 0.154 nm. The scanning speed was 4°/min, the scanning range was 20–80°, and the voltage and current were 40 kV and 400 mA, respectively. The microstructure and element distribution was observed and analyzed by field emission scanning electron microscopy, electron backscatter diffraction, and band energy spectrometer (SEM-EBSD-EDS) (SS-550 and JEOL JSM-4800 field emission scanning electron microscopes, Japan). The density was measured by Archimedes drainage method. The compression performance test is carried out on Shimadzu AG-X universal material testing machine, and the size of the compression sample is Φ four × 10 mm. In vitro mineralization experiments were conducted in simulated artificial body fluid (SBF) at 37 °C. The contents of each component of SBF were: 1 L deionized water, NaCl 8.00 g, CaCl_2_ 0.14 g, KCl 0.40 g, NaHCO_3_ 0.35 g, MgCl_2_·H_2_O 0.1g, Na_2_HPO_4_·12H_2_O 0.12 g, KH_2_PO_4_ 0.06 g, MgSO_4_·7H_2_O 0.1 g, and glucose 1 g. The mineralization experiment cycle is 5 days, with SBF solution replaced once a day. After soaking for 5 days, the surface morphology of the composite material was observed using scanning electron microscopy.

## 3. Results and Analysis

Figure 3 shows the relative density (*RD*) of Ti-27Nb-17Ta-8Zr/HA biocomposites with different HA contents. It can be seen that when HA content is 0%, 2%, 4%, 6%, 8%, and 10%, the relative densities of the composite materials are 98.6%, 97.9%, 96.8%, 96.1%, 93.9%, and 92.9%, respectively. The trend of changes in the figure indicates that as the HA content increases, the relative density of the composite material will decrease. When the HA content exceeds 6%, the relative density of the material will significantly decrease. The increase in HA content will lead to a significant decrease in the relative density of the composite material, mainly due to titanium (8.4 × 10^−6^/°C) and HA (15 × 10^−6^/°C) being too large, which leads to the inconsistency of shrinkage and porosity during the cooling process of SPS sintering. On the other hand, HA decomposes and reacts with Ti, as shown in Formulas (1)–(5) [17]. The reaction generates a small amount of gas, further increasing the surface pores of the material. As the HA content increases, the impact of these two aspects becomes greater with more and more pores, leading to a significant decrease in relative density.
(2)$$Ca_{10}(PO_{4}{)}_{6}{\left(OH\right)}_{2}\to Ca_{10}(PO_{4}{)}_{6}{\left(OH\right)}_{2-x}+xH_{2}O(g)$$
(3)$$Ti+2H_{2}O\to TiO_{2}+2H_{2}(g)$$
(4)$$Ca_{10}(PO_{4}{)}_{6}{\left(OH\right)}_{2}+TiO_{2}\to 3Ca_{3}(PO_{4}{)}_{2}+CaTiO_{3}+H_{2}O(g)$$
(5)$$Ca_{3}(PO_{4}{)}_{2}+\frac{1}{2}O_{2}+3Ti\to 3CaTiO_{3}+2P$$
(6)$$xTi+yP\to Ti_{x}P_{y}$$

Figure 4 shows the Brinell hardness of Ti-27Nb-17Ta-8Zr/HA composites with different HA contents. The average Brinell hardness of the Ti-27Nb-17Ta-8Zr/HA composite was 262, 282, 313, 317, 323, and 326 HBW, respectively. After adding HA, the hardness of the composites increased significantly. Before the HA content was 4%, the Brinell hardness showed a straight upward trend. When the HA content is greater than 6%, the hardness of the composite material gradually increases and tends to flatten out. The change trend of microhardness is a microscopic characterization. HA in composite materials is a hard phase, and the higher the content of HA in the composite material, the more HA is contained in a single grain, which will become harder and harder. Therefore, the hardness has been showing to be an increasing trend. The impact of HA on the microhardness of composite materials can be analyzed from the following three aspects. Firstly, the presence of HA can lead to solid solution strengthening, with some precipitated phases located at grain boundaries or phase boundaries, which not only serve as dispersion strengthening but also hinder grain growth in the material, thereby increasing the hardness of the material. Secondly, a small amount of HA reacts with elements such as titanium, resulting in certain changes in crystal structure, increased metallurgical bonding strength, and increased hardness in the material area. Thirdly, it can react with certain impurity elements to generate compounds, purify the interface between ceramic phase metal phase and ceramic phase ceramic phase, improve the material bonding strength, increase the resistance during crack propagation, and, thus, increase hardness. For the previously mentioned compressive strength, it is a macroscopic mechanical property. When an appropriate amount of HA is added, the β-Ti phase, while under the action of HA, the grain size has been refined to a certain extent, achieving good comprehensive mechanical properties. However, when the amount of HA added is too high, various defects and pores will occur, and the microhardness of the composite material will be affected.

The XRD analysis of Ti-27Nb-17Ta-8Zr/HA composites with different HA contents is shown in Figure 5. From the figure, it can be seen that the Ti-27Nb-17Ta-8Zr alloy is mainly composed of the β-Ti phase (JCPDS No. 65-5970) and a small amount α-Ti phase (JCPDS No. 65-3362). After adding HA, the composite material exhibited diffraction peaks of the HA phase (JCPDS No. 09-0432), and the diffraction peaks of the β-Ti phase gradually increased with the increase of HA content. Meanwhile, the diffraction peak intensity of the α-Ti phase decreases with increasing HA content. The strength of the α-Ti phase will increase with the increase of HA content. When the HA content is 10%, the diffraction peak intensity of α-Ti phase exceeds β-Ti phase further which proves that the higher the HA content, the more content of α-Ti phase. At the same time, it can also be observed that when the HA content exceeds 8%, some weak impurity peaks can be observed from the XRD spectrum, mainly composed of ceramic compounds such as CaTiO_3_ (JCPDS No. 77-0182), Ti_3_P_5_ (JCPDS No. 73-1319), and Ti_2_O (JCPDS No. 73-1570), indicating that HA undergoes decomposition or reaction during the sintering process.

In the early stage of SPS, the introduction of pulse DC current will produce instantaneous high temperature at the part of the powder contact (under the contact area, the resistance value is large). However, because the pulse DC current is high-frequency on-off, high temperature is generated when it is turned on. When it is disconnected, the temperature will quickly spread to the surrounding low temperature area, and HA has no time to decompose in such a short time. Due to the instantaneous high temperature, the particle energy increases, diffuses to the surrounding movement, and the pulse DC current will produce the effect of the electric field, driving the particle detour. When the temperature is gradually close to the sintering temperature, it will cause the decomposition of HA, resulting in CaTiO_3_, Ti_3_P_5_, and Ti_2_O.

The XRD analysis results indicate that the addition of the HA ceramic phase not only suppresses the generation of the β-Ti phase, it also promotes the transformation of β-Ti to the α-Ti phase. On the one hand, small rod-shaped HA is uniformly distributed on the surface of metal particles, which hinders the diffusion between alloy elements Ti, Ta, Nb, and Zr during the sintering process, resulting in the effect of phase transition elements Nb and Ta decreases and suppresses α-Ti to β-Ti transition. On the other hand, HA contains a large amount of interstitial element O, which is a type of titanium alloy α phase transition elements with improved β phase transition temperature. As the O element increases, the β phase transition temperature will significantly increase, and the presence of O element will also lead to β opposite direction to α phase transformation, generating more α-Ti phase [18]. After adding HA, some reactions will occur, generating a small amount of ceramic phase. This is mainly due to the decomposition of HA at high temperatures and the reaction between the decomposition products and Ti, which has also been reported in other studies [19]. The specific reaction equations are shown in Formulas (1)–(5) above.

SEM pictures of Ti-27Nb-17Ta-8Zr/HA biocomposites with different HA contents are shown in Figure 6. From Figure 6a, it can be seen that the Ti-27Nb-17Ta-8Zr alloy is mainly composed of gray, light gray, and white areas. Combining Figure 5 and EDS results, it can be seen that the gray area is composed of pure Ti; the light gray is Nb; and the white area is Ta and Zr. There are diffusion transition zones between different areas, mainly composed of β phase oriented. After adding ceramic phase HA, it was distributed on the β grain boundaries. Moreover, with the increase of HA content, needle-like and strip-like structures appear in the composite material of α-Ti phase; there are more and more pores in the composite material. When the HA content is 8%, the diffusion between metal particles is significantly weakened. When the HA content increases to 10%, a large number of microcracks appear inside the Ti particles. When the HA content is too high, more decomposition will occur, reacting with Ti, and the generated small amount of gas will further increase the porosity. Although the produced ceramic phase has certain biological activity, it will hinder the diffusion between metal elements and reduce the overall performance of the composite material. Therefore, the content of HA should be controlled.

Figure 7 and Figure 8 show the distribution of surface-scanning elements at HA content of 2% and 4%, respectively. It can be seen from the figure that the distribution of Ti, Nb, Ta, Zr, and other metal elements in the composite is relatively uniform; a solid solution zone has been formed in some areas, and the distribution of HA is relatively uniform. Some HA has segregation and is concentrated in the pores. As the HA content increases to 6%, HA segregation is evident. According to the analysis results of EDS composition, the element ratio of Ca:P:O is close to 5:3:13, indicating that the proportion of Ca, P, and O in the basic composite HA does not undergo decomposition or reaction.

Figure 9 shows the EBSD photos of Ti-27Nb-17Ta-8Zr/HA biocomposites with different HA contents. It can be seen from the figure that there are many small grains around larger grains such as Ti and Nb, which are mainly composed of Ta and Zr. As the HA content increases, the proportion of small grains increases, which means that the addition of HA hinders the diffusion reaction between Ta, Zr, Ti, and Nb, which corresponds to the results of SEM.

Figure 10 shows the compressive and yield strength of Ti-27Nb-17Ta-8Zr/HA composite materials with different HA contents prepared by SPS sintering. It can be seen that the tensile strength and yield strength show a trend of first increasing, then decreasing, and then slowly increasing with the addition of HA. When the HA content is 6%, the tensile and yield strength reach the lowest, reaching 610.5 MPa and 558.35 MPa, respectively, lower than the 872.5 MPa and 451.1 MPa of the Ti-27Nb-17Ta-8Zr alloy. The highest values of the Ti-27Nb-17Ta-8Zr/HA composite material (when HA content is 2%) are 909.5 MPa and 677.4 MPa, respectively. When the content of HA changes, there is a close relationship between the strength of composite materials and the microstructure evolution of HA. The differences in mechanical properties of composite materials are mainly caused by four aspects: the decrease in density, the reinforcement factor of nanoparticles, the factor of phase transformation, and the influence of reaction products. With the increase of HA content, although the alloy density slightly decreases, the strengthening and phase transformation effect of nano HA particles is obvious, leading to a certain increase in the tensile strength and yield strength of the composite material. As the HA content further increases, the relative density continues to decrease, while the phase transformation and reaction products significantly increase, resulting in a significant decrease in tensile strength and yield strength. However, with a further increase in HA content, the strengthening effect of nano HA particles is obvious. Although the relative density continues to decrease and microcracks appear, the tensile strength and yield strength of the composite material still slightly increase, which is consistent with the above detection and analysis results.

Figure 11 shows the elastic modulus of Ti-27Nb-17Ta-8Zr/HA composite materials with different HA contents prepared by SPS sintering. It can be seen that the elastic modulus of the prepared samples is all less than 20 GPa, close to the elastic modulus of human bone [20,21]. As the HA content increases, the elastic modulus shows a gradually increasing trend, reaching a maximum of 17.9 GPa when the HA content reaches 6%. However, the pattern of change is not obvious as the HA content changes.

Figure 12 shows SEM pictures of the compression fracture surface of Ti-27Nb-17Ta-8Zr/HA composite materials with different HA contents prepared by SPS sintering. From Figure 12a, it can be seen that there is only a small number of unmelted particles in the alloy, which also explains the reason for the high tensile strength and low yield strength of the alloy. As the HA content increases, small particles can be clearly observed at the fracture surface. In addition to unreacted HA, it should also be a new phase generated by the reaction. The higher the HA content, the more small particles there are. At the same time, unmelted large particles appear. This is related to HA and the new phase hindrance reaction; the results correspond to the mechanical performance tests. Compared to titanium alloy, ceramics are harder. During compression, they first slide relative to each other and accumulate the defects. At this time, ceramics have a shear effect relative to titanium alloy. Titanium alloy first breaks, and then microcracks appear. These are easily propagated at the weak hole walls, leading to interconnected weak hole walls, followed by macroscopic cracks near the pores. This leads to material failure. After the increase of HA content, there are more pores and relatively weaker pore walls. When the stress reaches the peak of the near linear elastic deformation stage, a large number of brittle pores are crushed layer by layer, leading to the material being more prone to failure.

The basic purpose of adding hydroxyapatite into the Ti-27Nb-17Ta-8Zr alloy is to enhance the biological activity of materials [22]. The combination of biomedical materials with human natural tissues after implantation is based on molecular and cellular levels. As a hard bone tissue replacement and repair material, the deposition and growth of bone, such as apatite on the surface of the implant material, is the premise of bone bonding between the implant material and natural bone tissue. Therefore, the in vitro mineralization ability of implanted materials is one of the criteria for evaluating their biological activity [23,24].

Figure 13 shows SEM pictures of the products obtained from mineralization of composite materials with different HA contents after soaking in SBF for 1, 3, and 5 days. Figure 13(a1–a3) is an SEM photo of the sample without HA addition. From the figure, it can be seen that there are spherical particles in the sample, which is consistent with the results in Figure 12a above. On the first day of mineralization, some smaller particles grew on the surface of the spherical particles, as shown in Figure 13(a1). On the third day of mineralization, the number of particles growing on the surface of the spherical particle scaffold increased, as shown in Figure 13(a2). Its energy spectrum analysis was demonstrated in Figure 14. The main elements on the surface of the spherical particle were O, P, Ca, and Zr. These elements were evenly distributed and the color of the surface scan was clear and rich, while Ti, Ta, and Nb were dim, indicating that the mineralization product hydroxyapatite grew on the spherical Zr particles. The other three metal elements were uniformly sintered in the matrix. By the 5th day of mineralization, the substrate morphology of the composite material has been completely invisible, and its surface is completely covered by evenly grown lamellar hydroxyapatite, as shown in Figure 13(a3). The energy spectrum analysis of it is shown in Figure 15. The main elements of the lamellar surface scan are O, P, and Ca, which are evenly distributed, while Zr, Ti, Ta, and Nb are evenly distributed in the matrix. Its corresponding Table 1 also shows that the lamellar is mainly O, P, and Ca. The mineralized product on the surface of the composite is hydroxyapatite. Figure 13(b1–b3) is an SEM photo of a sample with a 2% HA addition. On the first day of mineralization, some small particles grew on the surface of the matrix material, as shown in Figure 13 (b1). Continuing to mineralize until the third day, the morphology of the product remains uniformly growing small particles, as shown in Figure 13(b2). On the 5th day of mineralization, the hydroxyapatite on the surface of the matrix material changes from small particles to lamellar morphology, as shown in Figure 13(b3). Its energy spectrum analysis is shown in Figure 16. The main elements of the small particles are O, P, and Ca, which are evenly distributed, and the color of the surface scan is clear and rich. Although the matrix elements Zr, Ti, Ta, and Nb can also be seen in the surface scan, it can be seen in Table 2 of the corresponding element content that Zr is relatively high, and there are few others. It is observed that the mineralized product is lamellar hydroxyapatite, which grows evenly on the surface of the matrix material. Figure 13(c1–c3) shows SEM photos of samples with 4% HA addition. In the first day of mineralization, some small coral-like particles grew on the surface of the matrix material, as shown in Figure 13(c1). Compared with 2% HA samples with the same mineralization days, this is a morphology where the small particles grew taller, indicating that the higher the HA content added to the matrix material, the more conducive it is to the growth of surface mineralization products. On the third day of mineralization, the morphology of the product has transformed into a sheet-like morphology, with small particle loads on top of the sheet-like morphology, as shown in Figure 13(c2). On the 5th day of mineralization, the surface of the flake-shaped product has been smooth and free from small particle loading, as shown in Figure 13(c3). Figure 13(d1–f3) reveals SEM photos of samples with 6%, 8%, and 10% HA addition, respectively. The growth pattern of mineralized products is that on the first day of mineralization; some small coral-like particles grow on the surface of the matrix material. On the 3rd day of mineralization, the particle morphology changes to a sheet-like morphology. On the 5th day of mineralization, the flakes grow into larger flakes. Further weighing was carried out on the above samples to calculate the increment of mineralized products. The mass change rate of mineralized products is shown in Figure 17. It can be seen that as the number of mineralized days in the sample increases, the number of mineralized products continues to increase, indicating that the matrix material has a good mineralizing performance. The same mineralization time is on the 5th day, and as the amount of HA added to the sample increases, it shows a pattern of first increasing and then decreasing, with 4% HA gaining the most weight.

## 4. Conclusions

Ti-27Nb-17Ta-8Zr/HA composites with different HA contents were prepared using SPS process. As the HA content increases, its density gradually decreases. When the HA content is 10%, the density is only 92.9%. However, the Brinell hardness showed a gradually increasing trend, reaching 326 HBW at the highest. The Ti-27Nb-17Ta-8Zr alloy is mainly composed of the β-Ti phase and a small amount of α-Ti phase composition. The addition of HA not only inhibits the generation of the β-Ti phase, it also promotes the transformation of β-Ti to α-Ti phase. The compressive strength and yield strength show a trend of first increasing, then decreasing, and then slowly increasing with the addition of HA. The elastic modulus of Ti-27Nb-17Ta-8Zr/HA composite material is all less than 20 GPa, and when HA is 6%, it reaches 17.9 GPa. The mineralization experiment results showed that after adding HA, its biological activity was significantly improved. After 5 days of mineralization, as the amount of HA added to the sample increased, it showed a pattern of first increasing and then decreasing, with 4% HA gaining the most weight.

## Figures and Tables

**Figure 1 materials-16-05095-f001:**
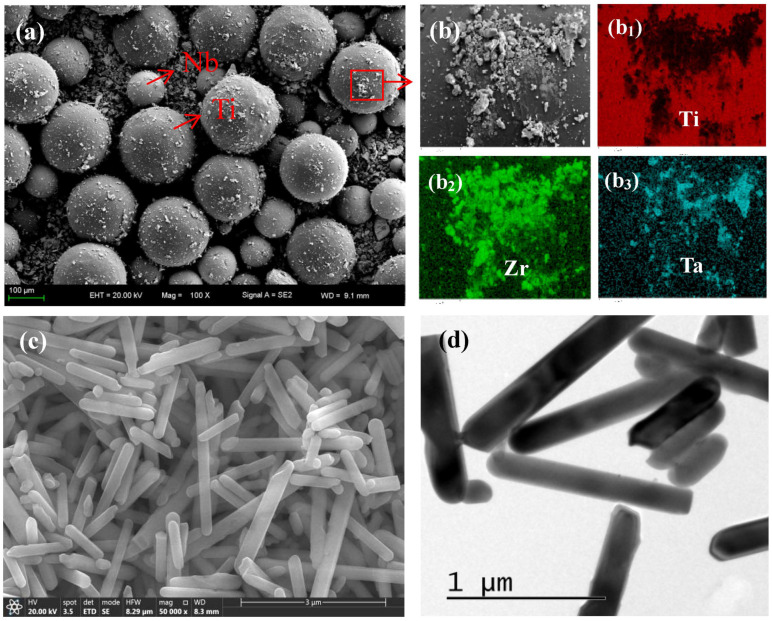
Morphology of powder. (**a**) SEM picture of Ti, Ta, Nb, and Zr mixed powders, and (**b**) Ti, Ta, and Zr mixed powders; (**b1**–**b3**) Elements mapping of Ti, Zr and Ta; (**c**) SEM photo of HA powders; (**d**) TEM photo of HA powders.

**Figure 2 materials-16-05095-f002:**
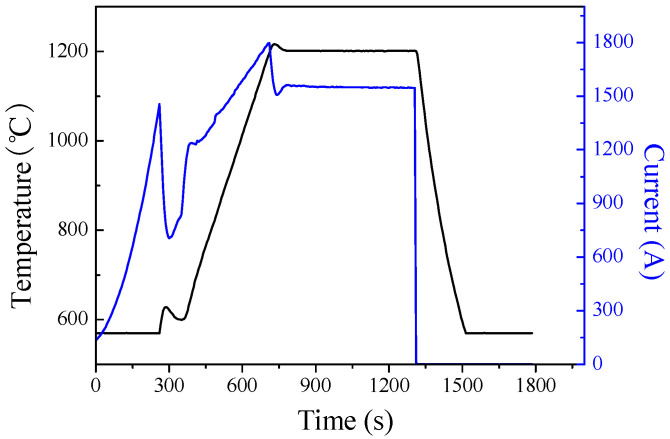
Process curve of SPS-sintered Ti-27Nb-17Ta-8Zr/HA composite material.

**Figure 3 materials-16-05095-f003:**
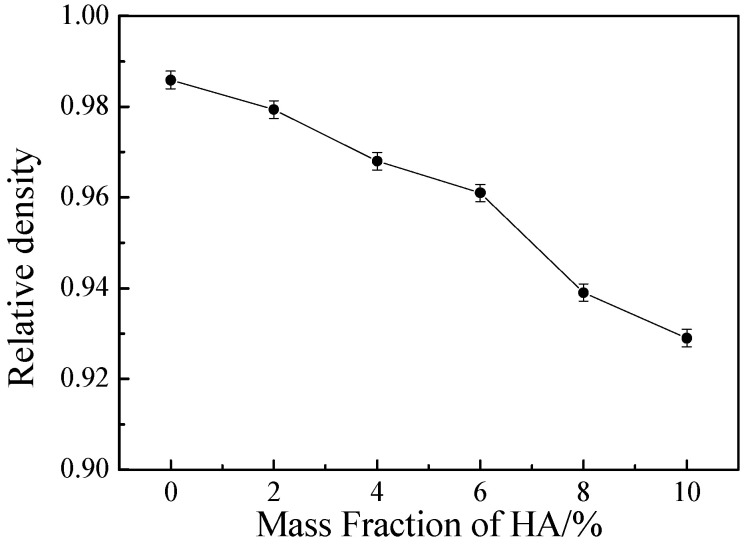
Relative density of Ti-27Nb-17Ta-8Zr/HA composite materials with different HA content prepared by SPS sintering.

**Figure 4 materials-16-05095-f004:**
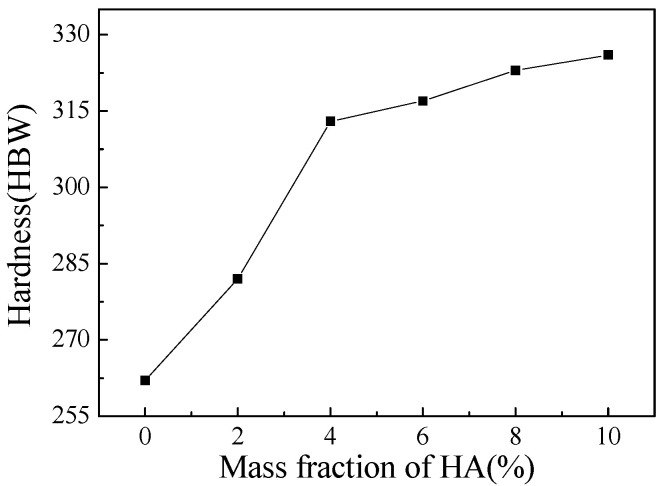
Brinell hardness of Ti-27Nb-17Ta-8Zr/HA composites with different HA content prepared by SPS sintering.

**Figure 5 materials-16-05095-f005:**
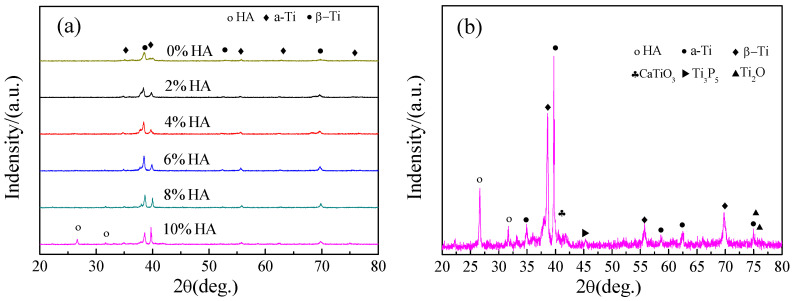
XRD analysis of Ti-27Nb-17Ta-8Zr/HA composite materials with different HA content (**a**) 0, 2%, 4%, 6%, 8%, and 10% HA; (**b**)10% HA.

**Figure 6 materials-16-05095-f006:**
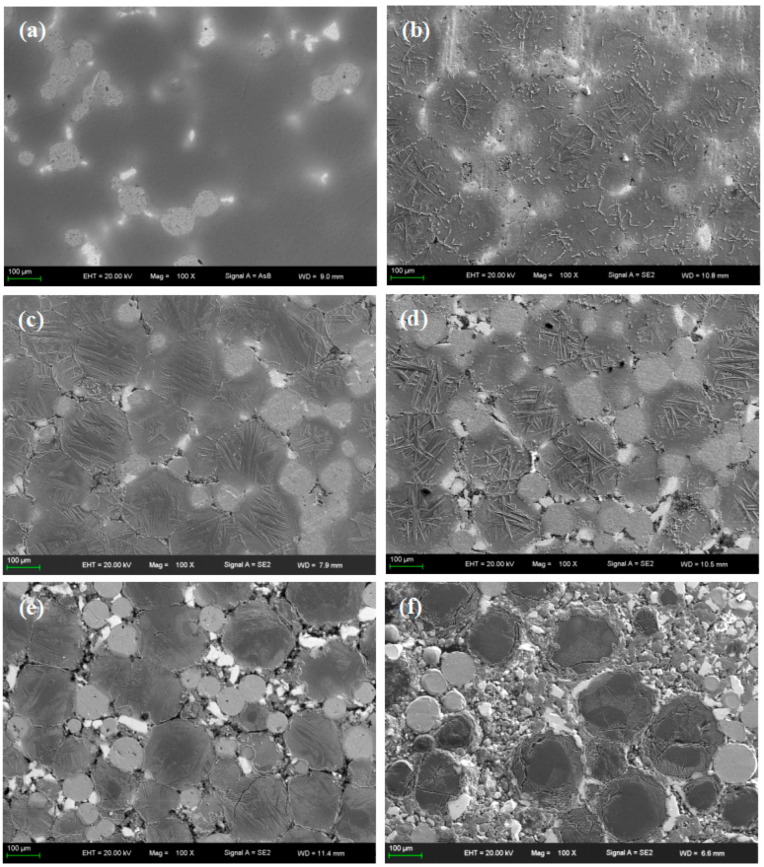
SEM pictures of Ti-27Nb-17Ta-8Zr/HA composites with different HA contents prepared by SPS sintering (**a**) 0; (**b**) 2% HA; (**c**) 4% HA; (**d**) 6% HA; (**e**) 8% HA; (**f**) 10% HA.

**Figure 7 materials-16-05095-f007:**
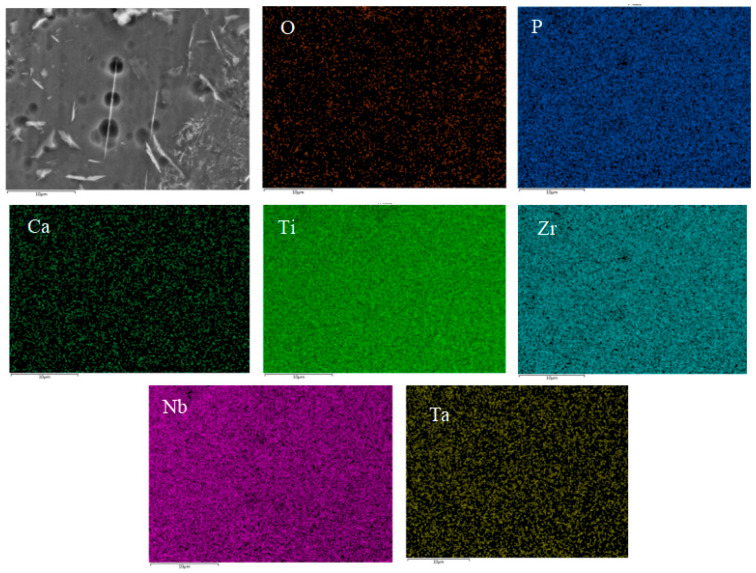
Elements mapping of Ti-27Nb-17Ta-8Zr/HA composite material with 2% HA content.

**Figure 8 materials-16-05095-f008:**
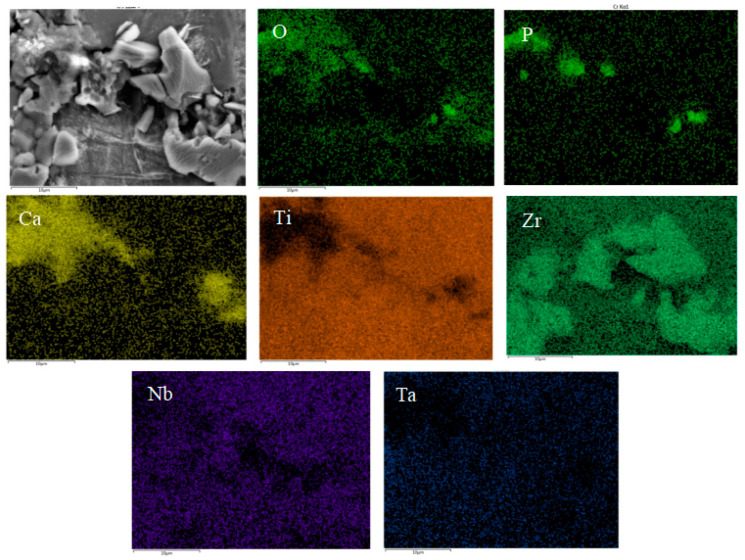
Elements mapping of Ti-27Nb-17Ta-8Zr/HA composite material with 10% HA content.

**Figure 9 materials-16-05095-f009:**
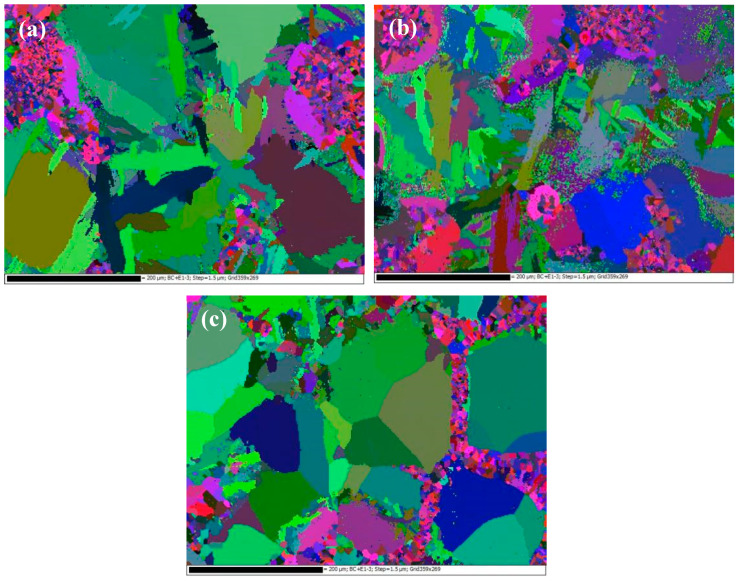
EBSD photos of Ti-27Nb-17Ta-8Zr/HA composites with different HA contents prepared by SPS sintering (**a**) 6% HA; (**b**) 8% HA; (**c**) 10% HA.

**Figure 10 materials-16-05095-f010:**
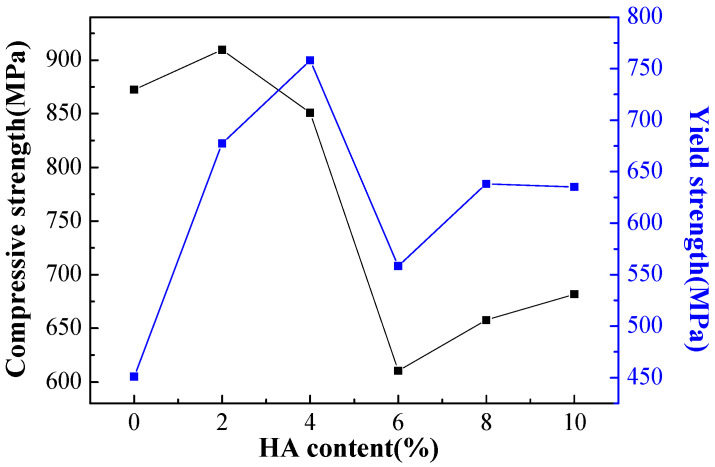
Compressive and yield strength of Ti-27Nb-17Ta-8Zr/HA composite materials with different HA content prepared by SPS sintering.

**Figure 11 materials-16-05095-f011:**
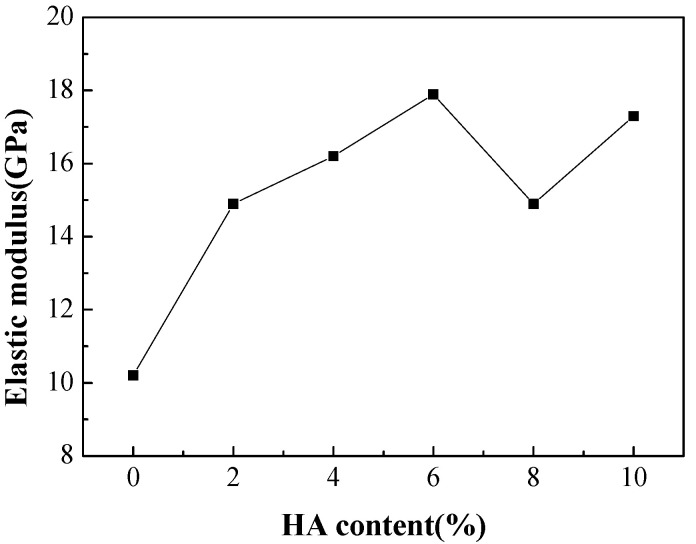
Elastic modulus of Ti-27Nb-17Ta-8Zr/HA composites with different HA contents prepared by SPS sintering.

**Figure 12 materials-16-05095-f012:**
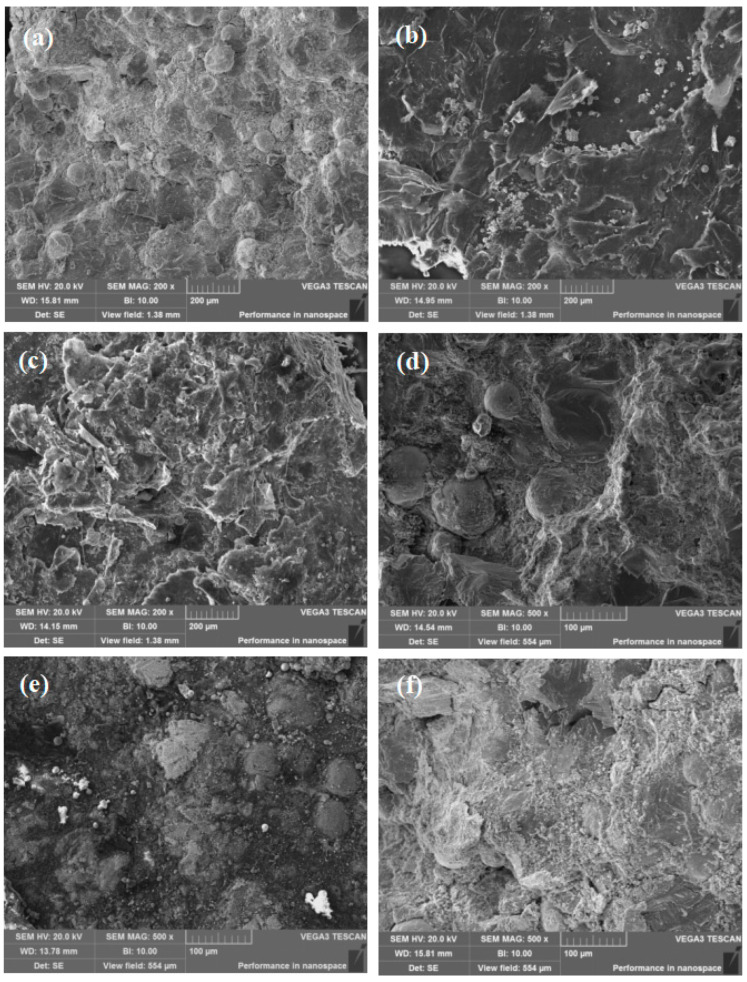
SEM pictures of compression fracture surface of Ti-27Nb-17Ta-8Zr/HA composite material with different HA contents prepared by SPS sintering. (**a**) 0; (**b**) 2% HA; (**c**) 4% HA; (**d**) 6% HA; (**e**) 8% HA; (**f**) 10% HA.

**Figure 13 materials-16-05095-f013:**
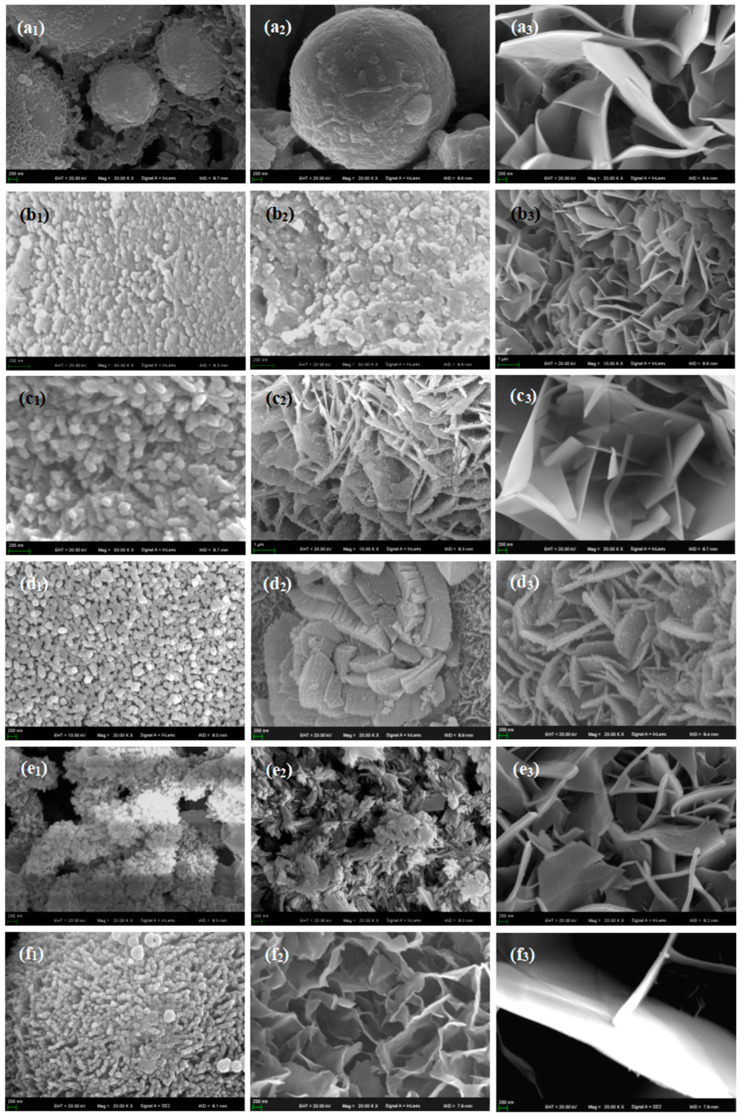
SEM pictures of material surface after immersion in artificial simulated body fluid for 1D, 3D, and 5D. (**a1**) 0 HA + 1D; (**a2**) 0% HA + 3D; (**a3**) 0% HA + 5D; (**b1**)2% HA + 1D; (**b2**) 2% HA + 3D; (**b3**) 2% HA + 5D; (**c1**) 4% HA + 1D; (**c2**) 4% HA + 3D; (**c3**) 4% HA + 5D; (**d1**) 6% HA + 1D; (**d2**) 6% HA + 3D; (**d3**) 6% HA + 5D; (**e1**) 8% HA + 1D; (**e2**) 8% HA + 3D; (**e3**) 8% HA + 5D; (**f1**) 10% HA + 1D; (**f2**) 10% HA + 3D; (**f3**) 10% HA + 5D.

**Figure 14 materials-16-05095-f014:**
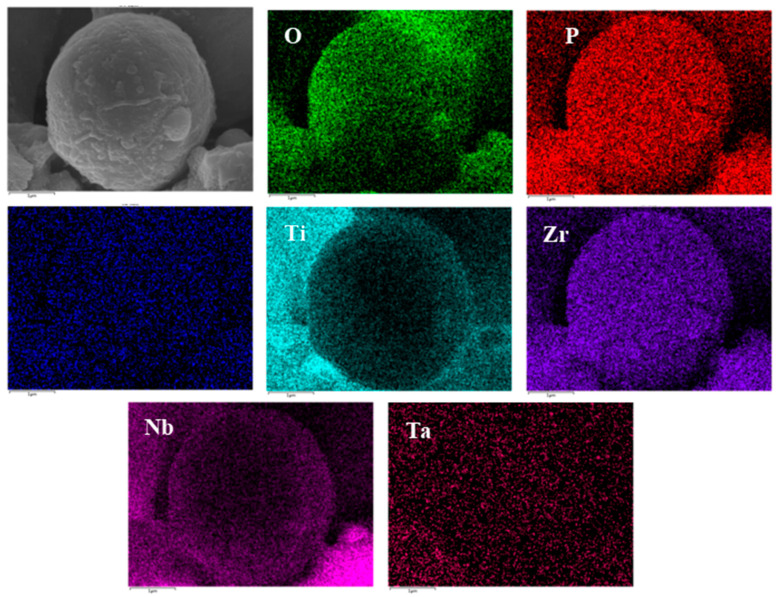
Elements mapping of 0% HA + 3D.

**Figure 15 materials-16-05095-f015:**
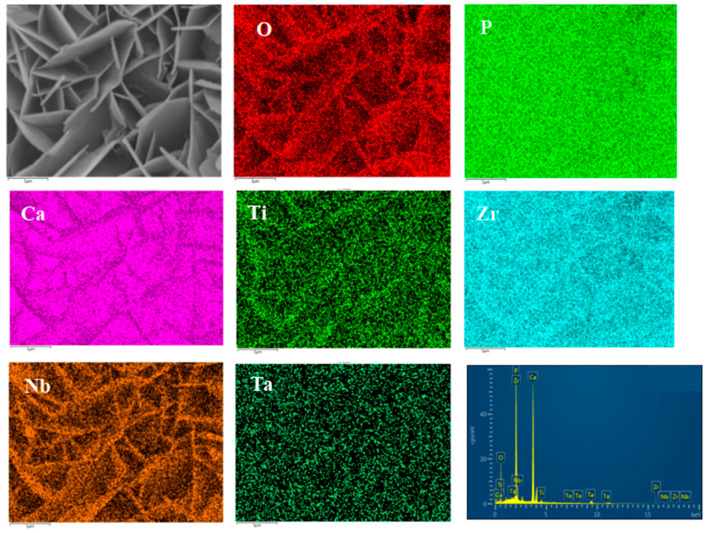
Elements mapping of 0% HA + 5D.

**Figure 16 materials-16-05095-f016:**
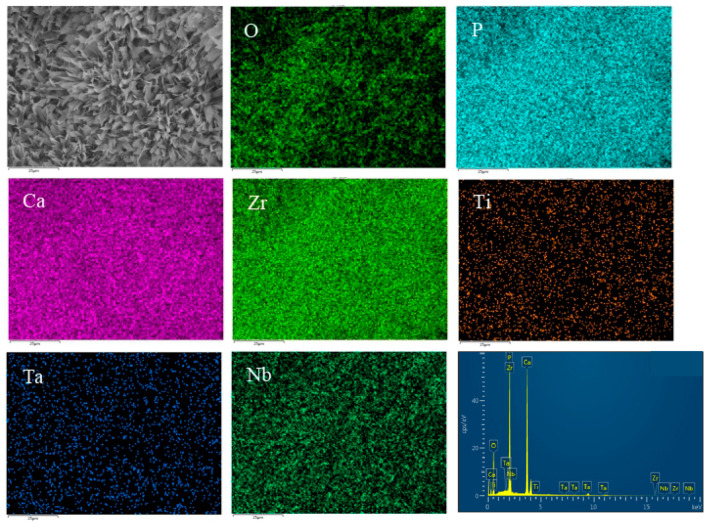
Elements mapping of 2% HA + 5D.

**Figure 17 materials-16-05095-f017:**
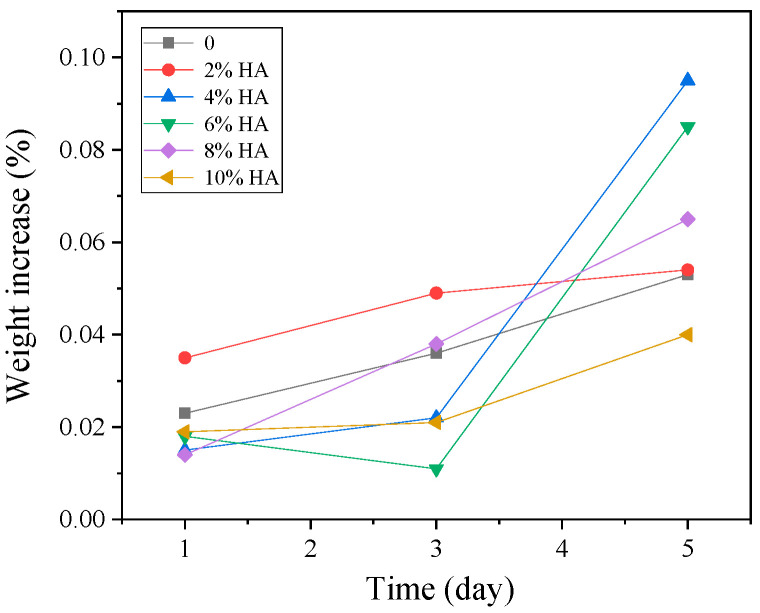
Mass change rate of mineralized products on materials soaked in artificial simulated body fluids for 1D, 3D, and 5D.

**Table 1 materials-16-05095-t001:** Distribution of elements corresponding to 0% HA + 5D samples.

Element	wt%	wt% Sigma	at% Sigma
O	36.77	0.30	59.81
P	17.30	0.14	14.54
Ca	33.70	0.21	21.88
Ti	1.12	0.05	0.61
Zr	11.07	0.30	3.16
Nb	0.00	0.00	0.00
Ta	0.03	0.23	0.00
Total	100.00		100.00

**Table 2 materials-16-05095-t002:** Distribution of Elements Corresponding to 2% HA + 5D Samples.

Element	wt%	wt% Sigma	at% Sigma
O	38.45	0.30	60.99
P	17.96	0.14	14.72
Ca	34.23	0.21	21.68
Ti	0.08	0.04	0.04
Zr	9.29	0.30	2.58
Nb	0.00	0.00	0.00
Ta	0.00	0.00	0.00
Total	100.00		100.00

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
