# Peer review of "Effect of HA Content on Microstructure and Properties of Ti-27Nb-17Ta-8Zr/HA Composite"

_materials, 2023, doi:10.3390/ma16145095_

Round 1
Reviewer 1 Report
The research aims to study the effects of different HA contents on the microstructure, evolution, mechanical properties and in vitro mineralization properties of Ti-27Nb-17Ta-8Zr/HA composites with different HA contents, prepared using SPS process. The microstructure and mechanical properties were also characterized by appropriate methods and techniques. As a result, convincing evidence of improved mechanical properties and biological activity was obtained for some of the compositions. Furthermore, it is shown that the addition of HA promotes the polymorphic transformation of beta-Ti to alfa-Ti. Some weaknesses of the work could also be noted:
1) Fig.1: (a) the figure caption is not correct
2) Figs. 7,8,15,…: better to use "elements mapping"
3) mandatory language revision is needed
4) some SEM/TEM micrographs could be saved. The large volume expands the work unnecessarily.
In conclusion, without shining with originality the work contains reasonable results and conclusions and could be published.

Author Response
Dear Dr. Sir,
Thanks for providing us with this great opportunity to submit a revised version of our manuscript. We appreciate the detailed and constructive comments provided by the reviewers. We have carefully revised the manuscript by incorporating all the suggestions by the review panel.
Reviewer 1.
1.Fig.1: (a) the figure caption is not correct
The figure caption of Fig.1: (a) has been changed. P4
- 7,8,15,…: better to use "elements mapping"
"elements mapping" has been used in Figs. 7,8,15,16. P12,20,21
- mandatory language revision is needed
The language revision has been done.
- some SEM/TEM micrographs could be saved.
The SEM/TEM micrographs had be saved.
Reviewer 2 Report
Dear authors, your work is interesting, but the following adjustments need to be made:
1. In my opinion, in the introduction, to increase readers' interest, it is necessary to add the advantages and disadvantages of the method used and compare it with techniques such as PEO, CVD, sol-gel, etc. I propose to add some new and interesting articles on this topic.
- https://doi.org/10.3390/ma15207374
- https://doi.org/10.3390/coatings10121249
- https://doi.org/10.3390/bioengineering7040127
2. At the end of the introduction, it is necessary to clarify the work's purpose and the analysis methods used.
3. P.2 It is necessary to add the countries and names of chemical reagent manufacturers and the equipment used. V-shaped mixer - manufacturer, country. SPS sintering system - ?. XRD, SEM -?
4. P.2 V-shaped mixer - you need to specify the mixing parameters in more detail. SPS sintering - explaining the powder processing procedure in more detail is necessary. Used currents, voltages, pulse frequency, type of die used, etc.
5. XRD - describe the type of X-ray tube used, currents and voltages, scanning angle, scanning speed, and scanning range.
6. Reactions 2 and 3 are not equalized correctly. Need to check.
7. Figure 5 Why are Ta, Nb, and Zr phases not visible on XRD plots? All phases found must be numbered from the ICDD library. It is also necessary to present a semi-quantitative phase analysis to compare the obtained results.
8. The above discussion about the effect of HA concentration on phase transformations is well presented but lacks evidence. Use literary sources to support some of the facts you present.
9. Figure 12 The discussion is descriptive. It needs to be expanded.
Author Response
Reviewer 2.
- In my opinion, in the introduction, to increase readers' interest, it is necessary to add the advantages and disadvantages of the method used and compare it with techniques such as PEO, CVD, sol-gel, etc. I propose to add some new and interesting articles on this topic.- https://doi.org/10.3390/ma15207374,ttps://doi.org/10.3390/ coatings10121249, https://doi.org/10.3390/bioengineering 7040127
The advantages and disadvantages of the method used and compare it with techniques have been added.P2-3
- At the end of the introduction, it is necessary to clarify the work's purpose and the analysis methods used.
The work's purpose and the analysis methods has been added.P3
- 2 It is necessary to add the countries and names of chemical reagent manufacturers and the equipment used. V-shaped mixer - manufacturer, country. SPS sintering system - ?. XRD, SEM -?
They all have been added.P3-4
- 2 V-shaped mixer - you need to specify the mixing parameters in more detail. SPS sintering - explaining the powder processing procedure in more detail is necessary. Used currents, voltages, pulse frequency, type of die used, etc.
They all have been added.P4
- XRD - describe the type of X-ray tube used, currents and voltages, scanning angle, scanning speed, and scanning range.
They all have been added.P5
- Reactions 2 and 3 are not equalized correctly. Need to check.
Reactions 2 and 3 have been modified.P6
- Figure 5 Why are Ta, Nb, and Zr phases not visible on XRD plots? All phases found must be numbered from the ICDD library. It is also necessary to present a semi-quantitative phase analysis to compare the obtained results.
It has been changed. P8-9
- The above discussion about the effect of HA concentration on phase transformations is well presented but lacks evidence. Use literary sources to support some of the facts you present.
Literary [19] is added to support some of the facts. P10
- Figure 12 The discussion is descriptive. It needs to be expanded.
The discussion of Figure 12 has been expanded.P15
Reviewer 3 Report
In this paper, Ti-27Nb-17Ta-8Zr/HAp-based composites were prepared by spark plasma sintering (SPS). A medical titanium alloy with good mechanical properties, wear resistance and corrosion resistance was combined with bioactive hydroxyapatite (HAp) ceramics with high biological activity and bone binding ability. The paper is definitely noteworthy and could be published in the journal "Materials", but requires a response to the following comments:
1. The abstract needs to be expanded to include the main numerical indicators of the material (yield strength, compressive strength, density, etc.).
2. Missing parameters to calculate the theoretical density. Please provide a formula how the authors arrived at the % - 10.1016/j.ijrmhm.2020.105385.
3. There is no understanding of the SPS process. It is better to extend the data on the kinetics of spark plasma sintering (first derivative of shrinkage) to know in which regimes the basic chemical reaction of transformation occurs (10.1016/j.jeurceramsoc.2022.02.007).
4. Figure 12 needs to be supported by EDX maps to understand how homogeneous the chemical interaction between the products is.
5. Figure 5 XRD analysis. It is necessary to recalculate the quantitative characteristics of the phases formed using the Rietveld method.
6. "Effect of HAp content on the microstructure and properties of Ti-27Nb-17Ta-8Zr/HAp composite" - In my opinion, the title does not reflect the essence of the paper. Firstly, it should be emphasised that there is the formation of reaction sintering. It is also shown that there is not only the formation of HAp, but also the accompanying phases of CaTiO3, Ti3P5 and Ti2O. Of course, this process occurs at 10% of the reaction mixture, so it is worth specifying at what concentrations the process of complete formation of hydroxyapatite occurs.
7. Perhaps the authors should emphasise the porosity of these materials. Is it necessary to have high density implants? Perhaps they should consider increasing the porosity of these materials with a "space holder" - 10.1016/j.ceramint.2014.09.045.
8. Page 12 Figure 9 shows the EBSD images of Ti-27Nb-17Ta-8Zr/HAp biocomposites with different HAp contents. It can be seen that there are many small grains around larger grains such as Ti and Nb, which are mainly composed of Ta and Zr. As the HAp content increases, the proportion of small grains increases, suggesting that the addition of HAp inhibits the diffusion reaction between Ta, Zr, Ti and Nb, which is consistent with the SEM results. In the absence of any reference or explanation, is this really the case?
9. All pages show SEM photos - it is advisable to replace them with - pictures.
10. Please provide a comparison table with modern biomaterials in use today and obtained by the SPS method (10.1134/S0036023620020138, 10.1016/j.pnsc.2019.07.004).
11. Why was it stored for exactly 5 days? (simulated body fluid for 5 days), is it enough to understand the processes, maybe it is worth to see the dynamics of the apatite layer growth with the replacement of the solution?
12. References not older than 5 years should prevail in the study, as well as a recommendation, it makes sense to pay attention to the authors to refer to MDPI journals.
Author Response
Reviewer 3.
- The abstract needs to be expanded to include the main numerical indicators of the material (yield strength, compressive strength, density, etc.).
The main numerical indicators of the material have been expanded.P2-3
- Missing parameters to calculate the theoretical density. Please provide a formula how the authors arrived at the % - 10.1016/j.ijrmhm.2020.105385.
The theoretical density is calculated.P5
- There is no understanding of the SPS process. It is better to extend the data on the kinetics of spark plasma sintering (first derivative of shrinkage) to know in which regimes the basic chemical reaction of transformation occurs (10.1016/j.jeurceramsoc.2022.02.007).
The SPS process has been added.P9
- Figure 12 needs to be supported by EDX maps to understand how homogeneous the chemical interaction between the products is.
The EDX maps have been done as seen in Figure 7 and 8. Moreover, the fracture surface is uneven and contaminated, so no EDX was performed
- Figure 5 XRD analysis. It is necessary to recalculate the quantitative characteristics of the phases formed using the Rietveld method.
The Rietveld method
There are significant differences in grain size, and there may be significant errors in quantitative analysis using XRD. For the sake of rigor, there is currently no analysis.
- "Effect of HAp content on the microstructure and properties of Ti-27Nb-17Ta-8Zr/HAp composite" - In my opinion, the title does not reflect the essence of the paper. Firstly, it should be emphasised that there is the formation of reaction sintering. It is also shown that there is not only the formation of HAp, but also the accompanying phases of CaTiO3, Ti3P5 and Ti2O. Of course, this process occurs at 10% of the reaction mixture, so it is worth specifying at what concentrations the process of complete formation of hydroxyapatite occurs.
The purpose of this article is to investigate the effect of HA content on tissue and performance, as decomposition occurs regardless of the amount of HA content. When the HA content is low, it is difficult to detect the low content of decomposition products. Increasing the content can characterize the decomposition products.
- Perhaps the authors should emphasise the porosity of these materials. Is it necessary to have high density implants? Perhaps they should consider increasing the porosity of these materials with a "space holder" - 10.1016/j.ceramint.2014.09.045.
Porous materials, which have a porosity greater than 40%, are also one of the implant materials. The porous materials of Ti-27Nb-17Ta-8Zr/HA are under research and will be reported in a specialized manner in the future.
- Page 12 Figure 9 shows the EBSD images of Ti-27Nb-17Ta-8Zr/HAp biocomposites with different HAp contents. It can be seen that there are many small grains around larger grains such as Ti and Nb, which are mainly composed of Ta and Zr. As the HAp content increases, the proportion of small grains increases, suggesting that the addition of HAp inhibits the diffusion reaction between Ta, Zr, Ti and Nb, which is consistent with the SEM results. In the absence of any reference or explanation, is this really the case?
The EBSD result is consistent with the SEM results. Moreover, this result has been validated in materials with similar compositions.
- All pages show SEM photos - it is advisable to replace them with - pictures.
They all have been replaced.
- Please provide a comparison table with modern biomaterials in use today and obtained by the SPS method (10.1134/S0036023620020138, 10.1016/j.pnsc.2019.07.004).
The biomaterials by the SPS method are not used and most of them are in the research stage.
- Why was it stored for exactly 5 days? (simulated body fluid for 5 days), is it enough to understand the processes, maybe it is worth to see the dynamics of the apatite layer growth with the replacement of the solution?
It is enough to test the data at high concentrations for 1,3 and 5 days.
- References not older than 5 years should prevail in the study, as well as a recommendation, it makes sense to pay attention to the authors to refer to MDPI journals.
The reference of MDPI journals has been prevailed. P23-24
Round 2
Reviewer 2 Report
Good job!
Reviewer 3 Report
Thanks to the authors for the work done on the responses to the comments! The paper can be accepted for publication in its current form.